# A 23-Year Observational Follow-Up Clinical Evaluation of Direct Posterior Composite Restorations

**DOI:** 10.3390/dj11030069

**Published:** 2023-03-01

**Authors:** Marie O. von Gehren, Stefan Rüttermann, Georgios E. Romanos, Eva Herrmann, Susanne Gerhardt-Szép

**Affiliations:** 1Department of Operative Dentistry, Dental School (Carolinum), Goethe University, Theodor-Stern-Kai 7, 60590 Frankfurt am Main, Germany; 2Department of Periodontics and Endodontics, School of Dental Medicine, Stony Brook University, 106 Rockland Hall, Stony Brook, NY 11794-8700, USA; 3Department of Oral Surgery and Implant Dentistry, Dental School (Carolinum), Goethe University, Theodor-Stern-Kai 7, 60590 Frankfurt am Main, Germany; 4Institute of Biostatistics and Mathematical Modelling, Goethe University, Theodor-Stern-Kai 7, 60590 Frankfurt am Main, Germany

**Keywords:** posterior composite restoration, clinical performance, long-term study, modified FDI criteria

## Abstract

The purpose of this observational follow-up clinical study was to observe the quality of posterior composite restorations more than 23 years after application. A total of 22 patients, 13 male and 9 female (mean age 66.1 years, range 50–84), with a total of 42 restorations attended the first and second follow-up examinations. The restorations were examined by one operator using modified FDI criteria. Statistical analysis was performed with the Wilcoxon Mann–Whitney U test and Wilcoxon exact matched-pairs test with a significance level of *p* = 0.05. Bonferroni–Holm with an adjusted significance level of alpha = 0.05 was applied. With the exception of approximal anatomical form, significantly worse scores were seen for six out of seven criteria at the second follow-up evaluation. There was no significant difference in the first and second follow-up evaluations in the grades of the restorations with regard to having been placed in the maxilla or mandible, as well as for one-surface or multiple-surface restorations. The approximal anatomical form showed significantly worse grades at the second follow-up when having been placed in molars. In conclusion, the study results show that significant differences regarding FDI criteria in posterior composite restorations occur after more than 23 years of service. Further studies with extended follow-up time and at regular and short time intervals are recommended.

## 1. Introduction

Since the development of resin-based composites, their popularity has grown over the years due to an increased demand for naturally colored dental restoration options in the posterior tooth region [1,2,3,4,5]. Decreasing acceptance of amalgam, growing concern about its biocompatibility, and issues with mercury have also made an alternative to amalgam necessary [6,7,8]. Less invasive cavity preparation, as a requirement for the insertion of direct composite restoration, and aesthetics are just some of the advantages of resin-based materials that make them the currently predominant material for dental restorations in numerous countries [9,10]. There is a broad selection of composites offered by manufacturers for direct dental restorations in anterior and posterior teeth. These composites vary from each other regarding their inorganic filler features, which influence the viscosity and the processing of the material [11], along with the physical properties [12,13], and hence affect the clinical performance of the restorations [14,15]. There have been reviews and analyses in various studies about the clinical performance of composite resins regarding the matrix composition and type, size, and content of different fillers [16]. In a 2011 review from Ferracane [16], nanohybrid and nano-filled composites were regarded as state-of-the-art concerning filler formulation in composite materials. In a different review from Alzraikat et al., from 2018, nanocomposites and hybrid composites showed comparable mechanical properties and clinical performance [17]. Clinical studies showed that nanohybrid composite resins performed similarly to conventional hybrid composite resins [18], and there was a similar performance of nano-filled and micro-hybrid composites [19].

There is a limited lifespan to dental restorations, as posterior composite restorations in a study by da Rosa Rodolpho et al. [14] showed a survival rate of 64% after 20 years; a study by van Dijken et al. [18] showed a success rate of 80.7% after 10 years; a study by Laegreid et al. [20] showed a survival rate of 87.7% after 3 years; a study by Opdam et al. [21] showed a survival rate of 84.7% after 12 years; and a study by van de Sande et al. [22] showed a survival rate of 69.9% after up to 15 years. Additionally, renewal “of a failed restoration leads to an increase in cavity size and destruction of tooth tissues” [23]. Consequently, increasing the longevity of restorations is of great significance in dentistry. Clinical evaluations of posterior restorations with composites provide an opportunity for knowledge of the material behavior and the results of the technique applied [14,18,20,21,22]. There are multiple factors that influence the longevity of restorations (for example, the operator, the selection of the materials, the clinical technique [24], and furthermore, the patients themselves, with their divergent oral hygiene and their different oral environments). The latter includes the location and extension of restorations, but bruxing habits and caries risks are also of considerable importance [5,15,25].

For the evaluation of dental restorations in long-term clinical studies, the USPHS (United States Public Health Service) criteria [26] were used by many researchers [5,27,28,29]. However, there was a need for a scale that was more discriminative and sensitive than the USPHS criteria in order to identify worsening and signs of failure in a timely manner. Hickel et al. suggested a new system in 2007 that was structured into three categories: aesthetic, functional, and biological [30]. To provide an opportunity for analysis and detailed description, each category was divided into different criteria and each criterion was evaluated according to a five-level rating of the restoration [31]. In 2007, the 16 criteria that were set by Hickel et al. were permitted by the Committee of the FDI World Dental Federation (Fédération Dentaire Internationale) [30] and in 2008 considered “Standard Criteria” [32]. However, the FDI criteria are not absolutely finalized and fixed; therefore, changes and/or adjustments are possible [32]. In addition, the examiners do not have to apply all 16 criteria but select the most appropriate ones [31].

There are several clinical [14,18,20,21,22,33,34,35,36] and systematic reviews [5,37,38] that have looked into the longevity and clinical effectiveness of posterior composite restorations, with evaluation periods that varied between 3 and 22 years. However, the quantity of long-term longitudinal clinical studies that have an evaluation span of more than 10 years is meager. Hence, the study at hand could be of interest as it evaluates posterior composite restorations more than 23 years after application.

During the years 1995–1996, adult patients attending the student courses of the Department of Operative Dentistry, Dental School (Carolinum), University Hospital, Goethe University Frankfurt (Germany), who were in need of posterior composite restorations, were informed of the study and invited to participate in the follow-ups. At a first follow-up, these restorations were then evaluated during a three-year observation period after application (1995–1997) at different intervals (6, 12, 18, 24, and 36 months) from different study groups. The results were published in 2016 [39]. In this present study, the same restoration group was given a follow-up examination after over 23 years.

The clinical evaluations of the direct restorations at the first and the second follow-up study were accomplished using the FDI criteria [30,32], which led to the ability to compare the results of the first and second follow-up examination with each other and allowed the comparison of the results with other international studies, such as that proposed by Marquillier et al. in 2018 [31]. A further benefit of the FDI criteria is that they are more sensitive to detecting dissimilarities in deciduous composite resin restorations [32] and led to better standardization of clinical judgment on restorations [31] than the USPHS criteria. In the present study, as in most comparable studies [40,41,42,43,44], because of the complex character of the FDI criteria, only certain criteria were selected according to the purpose of the respective research. This is consistent with the suggestions by Hickel et al. [32] and in line with a systematic review from 2018 about the usage of FDI criteria in clinical studies on direct dental restorations [31].

The goal of this second follow-up evaluation was to demonstrate changes in the quality of the examined FDI criteria over time. This is obviously dependent upon an investigator who should be seen as a unit of measurement of a subjectively descriptive nature. Consequently, the investigator always influences the measurements with a certain extent of subjectivity. To tackle this problem and to objectify the measurements to a certain degree, as well as to standardize the clinical judgment on the restorations as suggested by Hickel et al. [32], the experienced clinical dentist who conducted this study was trained and calibrated beforehand with a web-based and interactive training and calibration tool [45], which was still accessible online before starting the second follow-up examination, although 2019 was offline. Additional training and calibration on restored extracted teeth and on patients in a clinical setting were also carried out. Moreover, the dentist had no insight into the reached grades of the restorations in the first follow-up examination and had not initially placed the direct restorations. The aim of the study at hand was to assess the long-term clinical behavior as well as potential problems of composite restorations in the posterior area, more than 23 years after application and initial evaluation, based on modified FDI criteria. These problems can be avoided in the future by complying with an established clinical protocol and allowing a more accurate estimation of the clinical longevity of posterior composite restorations. Furthermore, the results should be used to confirm or refute existing knowledge.

The null hypotheses tested were that the scored grades at the second evaluation in 2019 would be the same as the scored grades at the first evaluation between 1995 and 1997. None of the posterior composite restorations were repaired or replaced; the tooth with the initial restoration either received prosthetic restoration or the tooth was lost. There was no difference between groups in scored grades regarding one-surface vs. multiple-surface restoration, placed in premolar vs. molar or maxilla vs. mandible.

## 2. Materials and Methods

Regarding the initial restorative procedures, all restorations were placed during the years 1995–1996 by dental students on courses in the Department of Operative Dentistry, Dental School (Carolinum), University Hospital, Goethe University Frankfurt (Germany). All the restorations were placed following a predefined clinical protocol “(for example: 1. no bevelling of the preparation; 2. consistent rubber dam application; 3. consistent metal matrix band application; 4. total-etch technique performed with Optibond FL (Kerr, Karlsruhe, Germany); 5. incremental composite application technique [with the composite Herculite XRV (Kerr, Karlsruhe, Germany)] in 2 mm layers each; and 6. surface contouring/finishing using carbide or fine diamond burs)” [39]. The rating and ranking procedures of the restorations at the first follow-up examination were executed by two trained dentists.

Prior to the start of the second follow-up examination, patients were provided with information on the study and signed written informed consent for inclusion in the present study. The study was conducted in accordance with the Declaration of Helsinki, and the study protocol was approved by the Ethics Committee of Goethe University Frankfurt (Ethical approval code: 130/18).

Patients that were selected for the present study (Figure 1) met the following inclusion criteria: Application of the posterior composite restoration happened in the 1995/1996 student courses of the Department of Operative Dentistry.All restorations were placed in permanent vital teeth that showed no pain symptoms at the time of placement.A micro-hybrid composite was used for the restorations (Herculite XRV, Kerr, Karlsruhe, Germany).The application followed a predefined clinical protocol.All patients were aged 18–70 years at the time of placement.All patients had been consecutively evaluated at the first follow-up examination between 1995 and 1997 within 36 months after the insertion of restorations.No night guard or similar device was provided to the patients.

Before the start of the follow-up examination, a case number calculation was performed based on the target variable “marginal adaptation”. For the main target variable “marginal adaptation”, a mean (standard deviation) score of 1.55 (in vivo at six months) was reported at baseline in a previous study [39], which was the starting point for the study at hand, and then increased to 2.25 (in vivo at 36 months). Values were based on a group size of 219 patients with 329 composite posterior restorations examined. The case number calculation was based on the same effect size and was based on the following assumptions: a significance level of alpha = 5% and a test power of 85%. The statistical analysis showed that if a total of 41 composite posterior restorations are followed up and evaluated, a statistical power of 85% can be achieved.

During the second follow-up examination between April 2019 and June 2019 (Figure 2), an average of 23 years and 5 months (standard deviation: 23 years, 5 months +/− 171 days) after initial placement of the direct restorations, all restorations were assessed according to FDI criteria [32]. Periapical radiographs, as advised by the FDI criteria [30] for the diagnosis of secondary caries, were not taken in order to protect patients from unnecessary radiation exposure, as approved by the Ethics Committee. Before the examination, the participants had to fill out a medical history form.

The evaluator, a blinded, trained, and calibrated dentist with clinical experience, used magnifying glasses with 2.7× magnification (starMed, Grafing, Germany), dental mirror (mouth mirror surface, DA026R, AESCULAP, Tuttlingen, Germany; mouth mirror handle, DA074R, AESCULAP, Tuttlingen, Germany), dental probe (silver line explorer probe, Hu-Friedy, Chicago, IL, USA), periodontometer (PCPUNC15, Hu-Friedy, Chicago, IL, USA), dental floss (Oral-B Essentialfloss, Procter & Gamble, Schwalbach am Taunus, Germany), and metallic matrix bands (Hawe Tofflemire Matrices, 0.05 mm, Kerr Hawe SA, Bioggio, Switzerland) for assessment of the restorations. Before evaluation, the surfaces of the teeth were dried with an air stream until fully dry. The experimental teeth were tested for pulp vitality by cold stimuli (Plurasol Kältespray, Pluradent, Offenbach, Germany) and for percussion. Furthermore, on the teeth with the restoration and on the corresponding reference teeth, the Papillary Bleeding Index (PBI) [46] and pocket depths on six sites per tooth (buccal, mesiobuccal, distobuccal, oral, mesiooral, and distooral) were measured. Intraoral photographs (EOS 450D, Canon, Tokyo, Japan) with a ring light (macro ring light MR-14EX, Canon, Tokyo, Japan) were taken of all examined composite restorations. There was documentation on standardized follow-up examination formulas for all data obtained by clinical means for this study. The second follow-up examination was not undertaken by the clinicians who at first placed the restorations in 1995–1996 nor by the dentists that executed the first follow-up examination.

The investigator did not “use the full set of the 16 [FDI] criteria but select[ed] the most suitable, according to the objectives of [this] study” [31]. The following seven FDI criteria were chosen for this study: 

Aesthetic properties:Surface luster;Aesthetic anatomical form;

Functional properties:3.Fracture of material and retention;4.Marginal adaptation;5.Occlusal contour and wear;6.Approximal anatomical form;

Biological properties:7.Tooth integrity (enamel crack, tooth fractures).

Each restoration was evaluated with these 7 FDI criteria and every criterion scored a grade according to a mutual six-step grade scale:1—clinically excellent/very good;2—clinically good;3—clinically sufficient/satisfactory;4—clinically unsatisfactory;5—clinically poor;6—restoration or tooth missing.

A restoration was considered a failure when the restoration scored a grade of 5 or 6 in any of the seven chosen FDI criteria. However, it was not categorized as a failure when there was the diagnosis of a new caries lesion on a surface that differed from the restored surface and therefore did not require the replacement of the analyzed restoration.

Each evaluated restoration was additionally classified in accordance with the ISO 3950 notation of the World Health Organization regarding the following criteria: premolar or molar, maxilla or mandible, and one-surface restoration or multiple-surface restoration.

To allow the comparison of the grading results of the first follow-up examinations from 1995 to 1997 with this follow-up study, the data were translated to a mutual six-step grade scale (as mentioned above) and only the abovementioned seven FDI criteria were compared. 

Statistical analysis was performed with SPSS Statistics 28 (SPSS Inc. an IBM Company, Chicago, IL, USA). If there was more than one evaluated restoration in the same patient, the results were considered independent. Two-sided tests with a significance level of alpha = 5% were applied. Differences between groups (premolar vs. molar; maxilla vs. mandible; one-surface restoration vs. multiple-surface restoration) were evaluated using the Wilcoxon Mann–Whitney U test and differences between paired observations on the first and second follow-up examination measurements on the same FDI criterion were evaluated with the Wilcoxon exact matched-pairs test. Bonferroni–Holm with an adjusted significance level of alpha = 0.05 was applied.

## 3. Results

The given grades of the 42 re-examined direct restorations of 22 patients at the first and second follow-up evaluations were used for the statistical analysis. The distributions of the seven evaluated FDI criteria were described by descriptive statistics. For each evaluated restoration, the statistical analysis for each of the FDI criteria was performed. Since the data at hand were not normally distributed, the Student’s *t*-test could not be applied to check for significant differences between the two evaluations. Hence, the Wilcoxon Mann–Whitney U test and the Wilcoxon exact matched-pairs test with a significance level of *p* = 0.05 were applied.

### 3.1. Clinical Performance in Accordance with Selected FDI Criteria

#### 3.1.1. Aesthetic Properties

Surface luster

At the first follow-up examination, only one restoration was found to have a clinically unsatisfactory score (grade 4: 2.4%). Most of the restorations showed clinically good (grade 2: 73.8%, *n* = 31) and excellent (grade 1: 23.8%, *n* = 10) results. The evaluated restorations at the second follow-up examination showed a broad range of grades: most of the evaluated restorations received a grade 2 (23.8%, *n* = 10), grade 3 (21.4%, *n* = 9), or grade 4 (14.3%, *n* = 6), and therefore no replacement of the restorations was necessary. There was only one restoration that showed luster comparable to enamel (grade 1: 2.4%). In total, 16 restorations could not be evaluated because the restoration or tooth was missing (grade 6: 38.1%).

The median and the interquartile range showed higher grades at the second follow-up examination (Figure 3). At the first follow-up examination, the median grade was 2.0 and the interquartile range was very small, at 0.25. At the second follow-up examination, the median increased to 4.0 and the interquartile range was remarkably larger, at 2.0.

There was no case in which the grade of the second follow-up examination was smaller than that of the first follow-up examination. In 32 cases, the grades at the second follow-up examination were greater than the grades at the first follow-up examination. In ten cases, there was a tie. A total of 42 direct restorations were evaluated. The criterion surface luster showed significant changes over time (*p* < 0.001; Wilcoxon exact matched-pairs test).

Aesthetic anatomical form

At the first follow-up examination, the criterion aesthetic anatomical form showed mainly clinically excellent (grade 1: 50.0%, *n* = 21) and clinically good (grade 2: 47.6%, *n* = 20) results, meaning that the anatomical form of the restorations was either ideal or only slightly deviated from the normal. However, 23 years later, at the second follow-up examination, only 7.1% (*n* = 3) of restorations scored a grade 1, and 16.7% (*n* = 7) scored a grade 2. A total of 16 restorations could not be evaluated because the restoration or tooth was missing (grade 6: 38.1%).

Both the median and the interquartile range of the results were higher at the second follow-up evaluation (Figure 3). At the first follow-up examination, the median grade was 1.5 and the interquartile range was 1.0. At the second follow-up examination, the median was considerably larger, at 4.0, as well as the interquartile range, at 3.25.

Only in one case was the grade of the second follow-up examination smaller than the grade at the first follow-up examination. In the vast majority, with 36 cases, the grades at the second follow-up examination were greater than the grades at the first follow-up examination (Figure 4). In five cases, there was a tie. A total of 42 direct restorations were evaluated. The criterion aesthetic anatomical form showed significant changes over time (*p* < 0.001; Wilcoxon exact matched-pairs test).

#### 3.1.2. Functional Properties

Fracture of material and retention

In total, 95.2% (*n* = 40) of the evaluated restorations at the first follow-up examination showed no fractures or cracks and therefore were evaluated at grade 1 (clinically excellent). The criterion fracture of material and retention was the best-rated criterion for the first follow-up examination. There was no restoration with a grade 2; two restorations (4.8%) showed “bulk fractures with partial loss” [32] of the restoration but affecting less than half of the restoration, and therefore scored a grade 4 (clinically unsatisfactory, but repairable). At the second follow-up examination, only 9.5% (*n* = 4) of the restorations still were evaluated with a grade 1. Notably, 16 restorations (38.1%) could not be evaluated because the restoration or tooth was missing (grade 6).

The median and the interquartile range showed higher grades at the second follow-up examination (Figure 3). At the first follow-up examination, the median grade was excellent, at 1.0, and the interquartile range was the smallest of all FDI criteria, at 0.00. At the second follow-up examination, the median of 3.5 and the interquartile range of 4.0 were remarkably larger.

There was no grade in the second follow-up examination that was smaller than in the first follow-up examination. In almost all cases (38 cases), the grades at the second follow-up examination were greater than the grades at the first follow-up examination (Figure 5). In four cases, there was a tie. A total of 42 direct restorations were evaluated. The criterion fracture of material and retention showed significant changes over time (*p* < 0.001; Wilcoxon exact matched-pairs test).

Marginal adaptation

At the first follow-up examination, only 7.1% (*n* = 3) of the direct restorations scored clinically unsatisfactory but reparable (grade 4) results. “Most of the fillings showed harmonious outline, no gaps, no white or discoloured lines” [32] (grade 1: 23.8%, *n* = 10) or only marginal gaps (<150 µm), small fractures removable by polishing, and slight ditching (grade 2: 69.0%, *n* = 29). It was noteworthy that at the second follow-up examination, none of the evaluated restorations received a grade 1, but showed a broad range of grades 2, 3, or 4, and therefore no replacement of the restorations was necessary. However, a total of 16 (38.1%) restorations could not be evaluated because the restoration or tooth was missing (grade 6).

The median and the interquartile range showed higher grades at the second follow-up examination (Figure 3). At the first follow-up examination, the median grade was 2.0 and the interquartile range was very small, at 0.25. At the second follow-up examination, the median of 4.0 and the interquartile range of 4.0 were remarkably larger.

There was no case in which the grade of the second follow-up examination was smaller than that of the first follow-up examination. In 32 cases, the grades at the second follow-up examination were greater than the grades at the first follow-up examination (Figure 6). In ten cases, there was a tie. A total of 42 direct restorations were evaluated. The criterion marginal adaptation showed significant changes over time (*p* < 0.001; Wilcoxon exact matched-pairs test).

Occlusal contour and wear

At 69.2% (*n* = 27), the vast majority of the evaluated restorations at the first follow-up examination showed physiological wear equivalent to enamel and therefore were evaluated with a grade 1 (clinically excellent). At the second follow-up examination, only 15.4% (*n* = 6) of the restorations were still evaluated with a grade 1, but 41.0% (*n* = 16) still showed physiological wear that differed only slightly from that of enamel and therefore received a grade 2. A total of 14 restorations (35.9%) could not be evaluated because the restoration or tooth was missing (grade 6).

The median and the interquartile range showed higher grades at the second follow-up examination (Figure 3). At the first follow-up examination, the median grade was excellent at 1.0, and the interquartile range was 2.0. At the second follow-up examination, the median was 2.0 and the interquartile range was considerably higher, at 4.0.

There was one grade of the second follow-up examination that was smaller than at the first follow-up examination. In 28 cases, the grades at the second follow-up examination were higher than the grades at the first follow-up examination. In ten cases, there was a tie (Figure 7). A total of 39 direct restorations were evaluated. The criterion occlusal contour and wear showed significant changes over time (*p* < 0.001; Wilcoxon exact matched-pairs test).

Approximal anatomical form

At the first follow-up examination, the criterion approximal anatomical form showed many clinically excellent results (grade 1: 44.0%, *n* = 11), meaning normal contact point and/or contour, but also many clinically unsatisfactory results (grade 4: 52.0%, *n* = 13), meaning a too-weak contact point and/or inadequate contour. The evaluated restorations at the second follow-up examination showed the full range of grades from one to six. In total, 32.0% (*n* = 8) of the restorations were grade 1, and 20.0% (*n* = 5) showed slightly too-strong contact but no disadvantage and/or a slightly deficient contour and were evaluated as grade 2 (clinically good). Seven restorations (28.0%) could not be evaluated because the restoration or tooth was missing (grade 6).

Both the median and the interquartile range scored higher results at the second follow-up evaluation (Figure 3). At the first follow-up examination, the median grade of 4.0 and the interquartile range of 3.0 were both considerably high. At the second follow-up examination, the median of 5.0 as well as the interquartile range of 4.25 were even larger. Of all criteria, approximal anatomical form showed the largest median and interquartile range at the second follow-up examination.

There were seven grades of the second follow-up examination that were smaller than at the first follow-up examination. In eleven cases, the grades at the second follow-up examination were higher than the grades at the first follow-up examination. In seven cases there was a tie. A total of 25 direct restorations were evaluated. As the only evaluated FDI criterion, the criterion approximal anatomical form showed no significant changes over time (*p* > 0.999; Wilcoxon exact matched-pairs test).

#### 3.1.3. Biological Properties

Tooth integrity (enamel cracks, tooth fractures)

At the first follow-up examination, only 2.4% (*n* = 1) of the restorations scored clinically unsatisfactory (grade 4) results. Most of the restorations showed complete integrity (grade 1: 47.6%, *n* = 20) or “small marginal enamel fracture (<150 µm), hairline crack in enamel (150 µm)” [32] (grade 2: 50.0%, *n* = 21). The evaluated restorations at the second follow-up examination showed the full range of grades from one to six. At 40.5% (*n* = 17), most of the evaluated restorations were evaluated with a grade 2. A total of 16 restorations (38.1%) could not be evaluated because the restoration or tooth was missing (grade 6).

The median and the interquartile range showed higher grades at the second follow-up examination (Figure 3). At the first follow-up examination, the median grade was 2.0 and the interquartile range was quite small, at 1.0. At the second follow-up examination, the median increased to 2.5 and the interquartile range was remarkably higher, at 4.0.

There was one grade of the second follow-up examination that was smaller than at the first follow-up examination. In 27 cases, the grades at the second follow-up examination were higher than the grades at the first follow-up examination. In 14 cases, there was a tie. A total of 42 direct restorations were evaluated. The criterion tooth integrity showed significant changes over time (*p* < 0.001; Wilcoxon exact matched-pairs test).

### 3.2. Comparison of Groups

There was no significant difference in either the first follow-up or in the second follow-up evaluation in the grades of the restorations with regard to having been placed in the maxilla or mandible (*p* > 0.05; Wilcoxon Mann–Whitney U test).

Regarding the grades of the restorations placed in a premolar or molar, there were no significant differences in any criteria at the first follow-up examinations (*p* > 0.05; Wilcoxon Mann–Whitney U test). However, one criterion showed significantly higher grades at the second follow-up examination when having been placed in molars: approximal anatomical form (*p* = 0.002; Wilcoxon Mann–Whitney U test). At the second evaluation, the median grade for the criterion approximal anatomical form for restorations in premolars was 2.0 and in molars 6.0. 

No significant differences were revealed in the grades if the restorations were one-surface restorations or multiple-surface restorations (*p* > 0.05; Wilcoxon Mann–Whitney U test), either in the first follow-up or in the second follow-up examination.

## 4. Discussion

The aim of the current study was the investigation of the long-term clinical behavior of and possible difficulties associated with posterior composite restorations in premolars and molars. It was carried out as an observational clinical follow-up of the research of Gerhardt-Szep et al. [39] and evaluated the same restorations with the same FDI criteria more than 23 years later, in order to be able to formulate more precise statements on the clinical performance of posterior composite restorations, compare the findings against each other, and therefore remove one of the limitations of the first follow-up study. There is a limitation regarding the generalizability of the results of this study due to the small sample size and the evaluation criteria based only on visual analysis. Moreover, the lack of standardization in the evaluation method is a limitation of this study. There is a broad range of ages in the selected study population, varying from 18 to 70 years at the time of placement of the direct composite restoration. The dietary habits of a person vary independently of age 

As there was a time interval of more than 23 years between these two follow-up studies, not all the initially evaluated patients and restorations could be re-examined, for diverse reasons, resulting in only a small case number of 22 patients with 42 corresponding restorations. Although this was in accordance with the statistical case number calculation, this must be seen as a limitation of this study, but is also a general problem of studies with longer evaluation periods, as also emphasized by a meta-analysis of prospective studies from 2015 [38]. The longer the evaluation period, the higher the frequency of dropouts [47], which leads to a deterioration in the power of the study as well as the clinical relevance. A further restriction of this study is that only the data of the first follow-up evaluation and, 23 years later, of the second follow-up evaluation could be compared. Hence, there lies a large time span between the two follow-ups without further evaluation. It was, therefore, not possible to monitor the reasons why and when the restoration was repaired or replaced or a tooth was lost in this period. For future examinations, it would be of great interest to reduce the interval between the next follow-up examinations to be able to better monitor the restorations over time, assess possible alterations, register the reasons for failure as in other studies [14,47,48,49], and record exactly when the failure occurred, in order to be able to describe the survival data in a Kaplan–Meier plot. The history of the restorations could not be investigated from the dental records of the university clinic, as the patients had all changed to different dental offices sometime after the first follow-up examination. This fact should also be considered regarding the survival of the restorations in the study at hand, as different studies stated that there is an increased chance of the replacement of dental restorations when patients change dentists [15,50,51,52,53,54].

Surface luster

Regarding the criterion surface luster, the restorations showed significantly worse grades at the second follow-up evaluation of this study. Although 16 restorations could not be evaluated and therefore scored a grade 6 because at the second follow-up, the restoration or the teeth was missing, it was striking that there was no restoration that scored a grade 5, which would have indicated that the restoration was in situ but clinically poor and required replacement. As long as the restoration was still in situ at the second follow-up, dull surfaces and multiple pores or even rough surfaces and voids could be removed by either repolishing or by reparation of the restoration. This is in line with Cieplik et al. [44], who recommended ensuring the longevity of the restorations and keeping a satisfactory surface luster. There should be a repolishing at each recall. 

Aesthetic anatomical form

The aesthetic anatomical form showed significantly worse grades at the second follow-up, and 16 restorations could not be evaluated and therefore scored grade 6 because the restoration or the tooth was missing. Only one restoration was rated as grade 5, which indicates a clinically poor condition that needs to be replaced. In total, 59.5% (*n* = 25) of posterior composite restorations at the second follow-up showed a broad variability in grades, between 1 and 4, which is in accordance with Hickel et al. [30] and a systematic review by Heintze et al. [55], which state that the clinical assessment of the criterion aesthetic anatomical form is to some degree subjective and hence is susceptible to a considerable amount of variability. This is particularly noticeable if the alteration of the posterior composite restoration is only to a minor extent. Therefore, care should be taken to ensure that in future studies, the investigator always has access to a photograph of the restoration at baseline and the follow-up recalls as a comparison.

Fracture of material and retention

In contrast to aesthetic anatomical form, the parameters of the criterion fracture of material and retention are straightforward to assess. The main reason for failure in posterior composite restorations reported in the literature [15,38,56] is the fracture of material and secondary caries. Moreover, it was stated that there was a higher association with the criterion fracture of material and retention in long-term studies that had over ten years of follow-up, according to retrospective studies after 17 [57] and 22 [14] years and one prospective clinical trial [34]. This goes in hand with the findings of this observational follow-up clinical study, as at the first follow-up, 95.2% (*n* = 40) of restorations scored a grade 1 in the criterion fracture of material and retention, but after more than 23 years, at the second follow-up, only 9.5% (*n* = 4) of restorations were still rated as grade 1, but 90.5% (*n* = 38) worsened in grade. In 2015, Pallesen et al. showed in a 30-year follow-up [56] that 70% of the restorations with material fractures appeared in patients that showed active parafunctional habits. Pallesen et al. [56,58] and other studies [5,59,60] concluded that the patient selection in the design of a clinical trial will have a high degree of influence over the results of the study and that excluding patients at high risk for secondary caries, bad oral hygiene, or parafunctional habits will result in far better longevity of restorative materials. The participants in the study at hand consisted of non-selected patients, which represented a general clinical patient population.

Marginal adaptation

The results of this observational follow-up clinical study regarding marginal adaptation show that at the second follow-up, in the majority of cases (26; 61.9%) a broad range of grades could be seen, including 2, 3, or 4, while no restoration was evaluated with a grade 1 or 5, and 16 (38.1%) restorations could not be evaluated because the restoration or tooth was missing (grade 6). Although the overall median grade of 2.0 at the first follow-up worsened to a median grade of 4.0 at the second follow-up, it should be noted that if the restoration was still in situ at the second follow-up, no replacement was necessary, but in the worst cases reparation at the margin of the restoration was required. In a randomized controlled study from 2015 by van Dijken et al. [47], there was no difference in the occurrence of cervical interfacial gap formation or even secondary caries when evaluating class II restorations after 15 years placed with either reduced shrinkage stress resin composite or mikrohybrid resin composite. As stated previously by Hickel et al. [30] and several other clinical findings, marginal adaptation may not only be associated with the restorative material but with a variety of influencing factors [47], such as the operator’s technique, caries risk, or bruxing activity [5,9,15,18,22,61]. The results of Loguercio et al. from 2019 [62] showed more suitable marginal adaptation and less marginal discoloration for posterior restorations bonded with etch-and-rinse adhesive than with self-etch adhesive, which is in agreement with a meta-analysis by Heintze et al. [37]. As in the study at hand, the etch-and-rinse strategy was performed when initially placing the restorations, which could be a reason for the sufficient and reparable restorations in 61.9% of cases at the second follow-up.

Occlusal contour and wear

The study results on occlusal contour and wear show many good results even after more than 23 years of clinical service, with 56.4% (*n* = 22) of restorations being grade 1 and 2 at the second follow-up. This is in line with Demarco et al., who stated that occlusal wear is seldom considered to be the reason for the failure of restoration and not a clinical problem of posterior composite restorations [15]. However, occlusal wear could be a problem for patients with parafunctional habits such as bruxing or clenching [4,15]. This is not always easy to measure for the examiner, as wear is usually not homogenous across all the restoration surfaces or if the whole occlusal surface was reconstructed. A comparison with the enamel of an unrestored reference tooth would be helpful, as well as a comparison with baseline and follow-up photographs of the restoration [30,32].

Approximal anatomical form

Of all the criteria in this clinical trial, the approximal anatomical form had the largest median (5.0) and the largest interquartile range (4.25) at the second follow-up evaluation, which clearly shows the difficulty in designing an anatomically correct form and an adequate approximal contact. However, it is necessary to point out, as this study includes class I and II restorations, that only class II restorations could be taken into account for this criterion. Additionally, the approximal anatomical form was the only criterion in this clinical trial that showed no significant changes over time (*p* > 0.999; Wilcoxon exact matched-pairs test). As the direct restorations of the study at hand were placed in the years 1995–1996, the results are similar to those of a clinical study from 1999, which stated that it was difficult for dentists to accomplish sufficient approximal contact [28]. The further development of techniques for creating a sufficient approximal contact has helped to eliminate this problem so that the operator can now choose between different matrices and separation ring systems [15]. However, it is not only the approximal contact that can be problematic. There could be an intact approximal contact of restoration; however, the approximal silhouette can be unsatisfactory, giving rise to plaque accumulation, which can lead to damage to the periodontal tissues or secondary caries [32]. Additionally, the approximal cervical restoration gap of class II cavities often lies in the dentin, which might provide an explanation for the bad grades of the approximal anatomical form [39]. In this area, the polymerization shrinkage stress in the composite layers may have more influence and could result in the formation of local interfacial gaps, causing microleakage [63,64]. As described in a systematic review and meta-analysis by da Veiga et al. in 2016 [65], it is still unsure whether this could lead to clinically relevant gap widths at the outer margin of the restoration [63]. The extent of this volumetric shrinkage is also influenced by other components such as resin matrix formulation, filler amount in the composite material, conversion rate [47], direct or indirect resin composite restoration [66], or use of an incremental restoration technique [67]. 

Tooth integrity (enamel cracks, tooth fractures)

Regarding tooth integrity, the results of the clinical trial showed an acceptable degradation in grade over time, as the median grade only worsened from 2.0 at the first follow-up to 2.5 at the second follow-up, pointing out that most of the examined restorations were evaluated with a grade 2 at the first follow-up (50.0%, *n* = 21), as well as at the second follow-up (40.5%, *n* = 17). This study can be roughly compared to that of Cieplik et al. from 2022 [44], which, similarly to our findings, noticed hairline cracks (grade 2) in approximately half of the restored teeth but also did point out that significant degradation in the criterion tooth integrity was not a major concern in their study.

Several studies have reported that the tooth type has a direct effect on the longevity of the restoration, with restorations placed in premolars showing significantly better performance than restorations placed in molars [5,14,34,49,57,60,68]. Palotie et al. [69] found in a 13-year period of observation that restorations in premolars experienced fewer failures than restorations in molars, with an annual failure rate of 3.1% and 5.2%, respectively. These findings could be explained by the knowledge that restorations in molars are exposed to greater occlusal forces than restorations placed in premolars [15,48]. It must be noted that although all restorations in this study were carried out on posterior teeth, the masticatory load varies even in the posterior region. Furthermore, it is suspected that more difficult access to the working field when restoring molars could also lead to worse results [48]. However, in the present study, only the criterion approximal anatomical form at the second follow-up showed a significantly better performance of restorations in premolars than in molars (*p* = 0.002; Wilcoxon Mann–Whitney U test), meaning that for all other criteria, significant differences were found neither at the first nor at the second follow-up. Additionally, several other studies also did not confirm that restorations in premolars performed significantly better than restorations in molars [47,48,56,58,65]. It is noteworthy that in this observational follow-up clinical trial, before Bonferroni–Holm with an adjusted significance level of alpha = 0.05 was applied, six out of seven FDI criteria at the second follow-up showed significant differences. This shows a trend pointing towards better-performing restorations in premolars than in molars before Bonferroni–Holm was applied. Applying Bonferroni–Holm to counteract the problem of multiple comparisons and control the family-wise error rate strongly influenced the outcome of this clinical trial, as it is a very conservative approach to data handling. Furthermore, because of the small sample size, the Bonferroni–Holm had a strong effect on our data, and the results may be more likely to indicate statistically significant differences if the sample size was higher, although the sample size was chosen according to a calculation based on previous clinical studies with similar design and in accordance with the Institute of Biostatistics and Mathematical Modeling Frankfurt. 

Considering the impact of the number of restored surfaces on the longevity of restoration, there are several studies showing significantly lower failure rates for single-surface than for multi-surface restorations in posterior teeth [5,14,57,59,62,70]. Van de Sande et al. stated in their clinical study about the survival of posterior composite restorations after 18 years that the number of restoration surfaces could favor the collapse of the restoration, primarily because of the less sound tooth structure that remains, therefore significantly affecting the longevity of restorations [49]. The clinical study at hand, however, found no significant difference in longevity regarding single- or multi-surface restorations after applying Bonferroni–Holm. There was even a tendency towards a better survival of multi-surface restorations before applying Bonferroni–Holm, as the criterion aesthetic anatomical form at the first follow-up and the criterion approximal anatomical form at the second follow-up showed significantly better results in multi-surface restorations. The reason for this is not clear and is not in line with the relevant literature. Furthermore, the longevity of direct restorations also depends on the cavity size and area of the restoration. It is a limitation of this study, that there was no evaluation of these parameters. Moreover, cavity type would influence the longevity of direct restorations. Although this observational follow-up clinical study did examine class I and class II restorations in their follow-ups, there was no differentiated evaluation of these data, which must be seen as a limitation of this study. Better standardization in the data collection would have been beneficial for the study outcomes.

There are inconsistent and scarce findings regarding the longevity of posterior composite resin restorations located in the maxilla or mandible. In a study of the longevity of multi-surface restorations in posterior teeth by Palotie et al. [69], it was found that restorations in the maxilla clearly outlasted restorations in the mandible. However, if looked at from a cariological perspective, posterior teeth in the maxilla and mandible are equivalent [71]. Lucarotti et al. [51] even mentioned a modest but significant distinction in terms of the benefit of restorations in the mandible, but it must be noted that the study combined posterior and anterior teeth and the data did not incorporate composites in molars. On the other hand, in a randomized controlled study by Pallesen et al. [56], restorations in the upper arch and in the lower arch showed a similar failure frequency. This is in line with the study at hand, which found no significant difference in the longevity of posterior composite restorations either at the first or at the second follow-up evaluation with regard to being placed in the maxilla or mandible after applying Bonferroni–Holm. 

The longevity of posterior direct composite restorations is affected by numerous risk factors related to the patient, operator, and material. In a systematic review and meta-analysis by Opdam et al. [5], the composite material type was not found to be a significant risk factor concerning the survival of a restoration. Demarco et al. concluded that in most posterior teeth, the quality standard of resin composite materials is adequate to accomplish the clinical requirements [15,72]. This is equally underlined by several studies showing good clinical results for posterior composite restorations with annual failure rates of 0.9% to 3.3% when using hybrid materials [5,25,37,56,58]. Patient-related factors such as parafunctional habits and caries risk significantly affect the longevity of a restoration [22,72], meaning that in the setup of a clinical trial, the output of the study is influenced to a high degree by patient selection [56,58]. The exclusion of patients with risk factors such as high caries activity, bruxism, or poor oral hygiene will result in clearly improved results regarding durability [5,59,60]. The sample of patients in the study at hand represented non-selected participants, thus representing a normal clinical patient population. The most important factors influencing the longevity of posterior composite restorations are probably those related to the operator [48,72], such as education, training level, and accuracy of work. Furthermore, it is the operator who decides if a restoration requires replacement. In diagnostic decisions, there is a high level of variability among dentists. Different studies of secondary data show that the chance of a restoration being replaced increases with the patient changing dentists [50,51,52,53], and for composite and amalgam restorations a reduced lifetime has been described for patients changing clinicians [54]. Moreover, practice-based studies from 2016 [73,74] show that the risk of restoration failure is twice as high between different dental offices and that there are substantial differences in the longevity of restorations among clinicians. To prevent the broad variability regarding diagnosis and treatment options among dentists, evidence-based reasons for the monitoring, refurbishment, repair, or replacement of a restoration with standardized criteria should be used [56,59]. As suggested by Hickel et al. [32], the international and standardized FDI criteria can be applied by dental students, general practitioners, and researchers in order to ensure quality, avoiding premature replacement. This is of great importance because the repair of a restoration is acknowledged as a conservative solution that preserves tooth structure, can be carried out at a lower cost, and demands less clinical time [14,15,75]. The increase in the lifespan of the population in Western countries and the simultaneously growing number of retained teeth may lead to an overload of restorations in need of intervention in the near future [69]. Just slightly increasing the longevity of restorations might result in substantial savings in treatment costs [76]. Therefore, future dentistry should use FDI criteria to lean in the direction of choosing more conservative treatments such as monitoring, refurbishment, or repair when handling defect restorations, in order to prevent over-early failures and improve the longevity of restorations.

## 5. Conclusions

In conclusion, the observational follow-up clinical study showed that after more than 23 years of clinical service, six out of seven FDI criteria at the second follow-up had significantly worse grades. Only the criterion approximal anatomical form at the second follow-up showed significantly better performance for restorations in premolars than in molars. Further studies with extended follow-up time and at regular and short time intervals are recommended.

## Figures and Tables

**Figure 1 dentistry-11-00069-f001:**
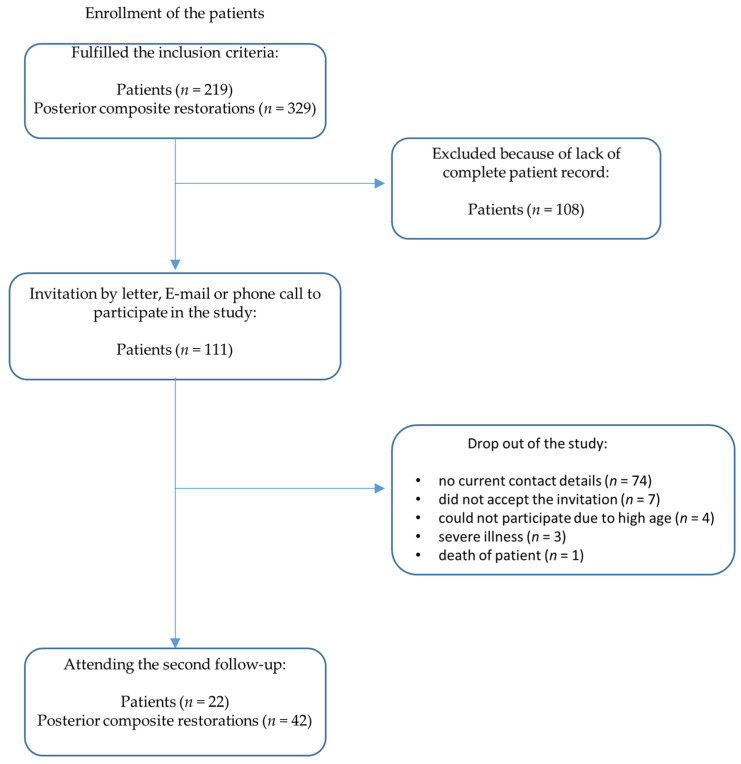
Flow diagram: enrollment of the patients.

**Figure 2 dentistry-11-00069-f002:**
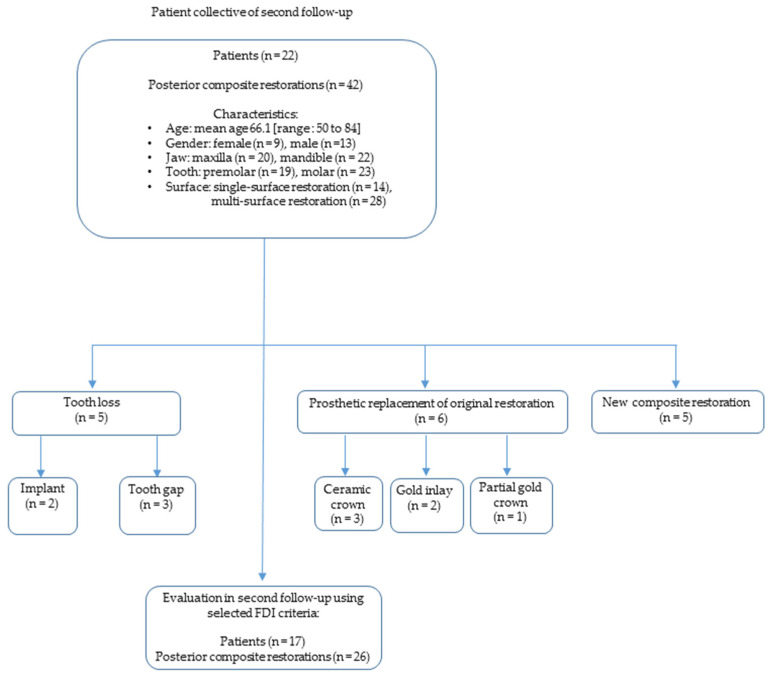
Flow diagram: patient collective of second follow-up.

**Figure 3 dentistry-11-00069-f003:**
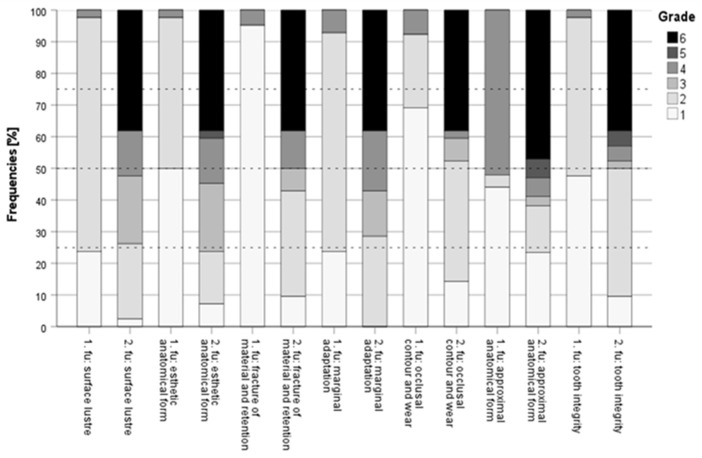
Stacked bar charts with median and first and third quartile for 22 patients and 42 restorations at the first follow-up (1.fu) and the second follow-up (2.fu) examination.

**Figure 4 dentistry-11-00069-f004:**
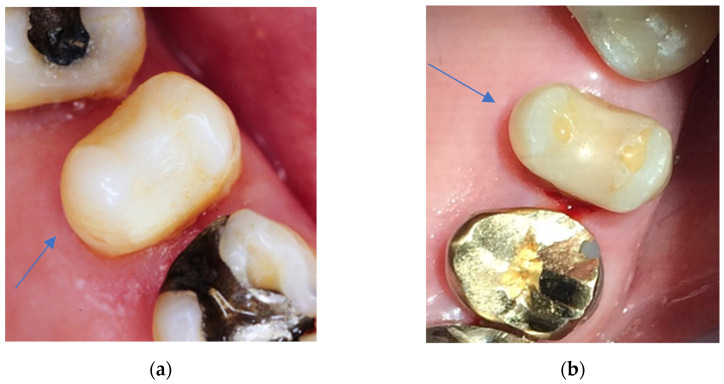
(**a**) Clinical view of a Herculite XRV restoration at first follow-up examination (25 mod). Twelve months after placement, the criteria anatomical form, marginal adaptation, tooth integrity, and surface luster did not score excellent but only good grades (grade 2). (**b**) Clinical view of the Herculite XRV restoration displayed in Figure 4a after 24 years in situ. Marginal fractures with exposed dentin are apparent; therefore, the restoration was clinically unsatisfactory (grade 4) in the criteria of marginal adaptation and had to be repaired. The aesthetic anatomical “form deviates from the norm but is aesthetically acceptable” [32] and the restoration shows a dull surface with multiple pores; therefore, anatomical form and surface luster scored only clinically sufficient/satisfactory results (grade 3).

**Figure 5 dentistry-11-00069-f005:**
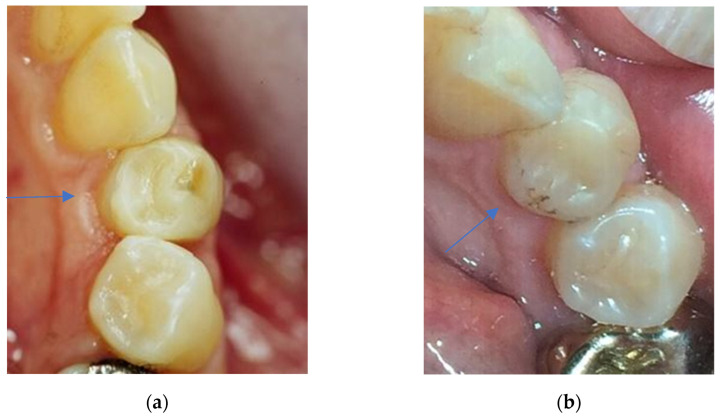
(**a**) Clinical view of a Herculite XRV restoration at first examination (34 modvl). Six out of seven criteria scored clinically excellent/very good (grade 1) results. The approximal anatomical form was the only criterion with a clinically unsatisfactory (grade 4) result. (**b**) Clinical view of the Herculite XRV restoration shown in Figure 5a after 24 years in situ. All criteria scored clearly higher grades in the second examination. Aesthetic anatomical form and retention received the highest grades (grade 4 = clinically unsatisfactory). Only the criteria approximal anatomical form scored lower grades in the second examination (grade 2 = clinically good) than in the first examination, due to the new prosthetic restoration on the adjacent tooth, which led to a new, more appropriate approximal contact point.

**Figure 6 dentistry-11-00069-f006:**
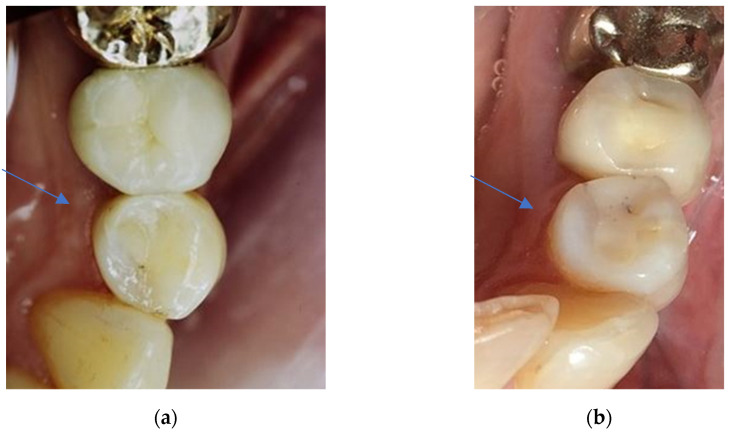
(**a**) Clinical view of a Herculite XRV restoration at first examination (34 od). Aesthetic anatomical form, approximal anatomical form, tooth integrity, and retention were grade 1 (clinically excellent/very good). Marginal adaptation, occlusal contour/wear, and surface luster were grade 2 (clinically good). (**b**) Clinical view of the Herculite XRV restoration displayed in Figure 6a after 24 years in situ. The grades of the criteria approximal anatomical form and occlusal contour/wear stayed the same, and the five other criteria were all evaluated to be of higher grades. Marginal adaptation scored the highest grade (grade 4 = clinically unsatisfactory). The biggest difference in grades between the first and the second evaluation was for the criteria marginal adaptation and retention.

**Figure 7 dentistry-11-00069-f007:**
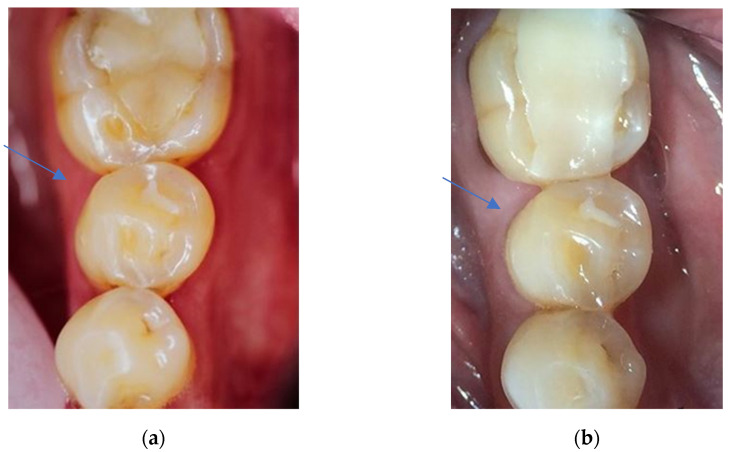
(**a**) Clinical view of a Herculite XRV restoration at first examination (35 od). Aesthetic anatomical form, marginal adaptation, tooth integrity, retention, and surface luster were examined with grade 1 (clinically excellent/very good). Occlusal contour/wear scored grade 2 (clinically good) at the first examination. Approximal anatomical form scored the highest grade of all criteria at first examination: grade 4 = clinically unsatisfactory. (**b**) Clinical view of the Herculite XRV restoration displayed in Figure 7a after 24 years in situ. Only one criterion, approximal anatomical form, scored a lower grade at the second evaluation (grade 3 = clinically sufficient/satisfactory) than at the first evaluation. This was the case because the restoration of the neighbor tooth was replaced, and thus a new, clinically sufficient approximal contact point was created. Occlusal contour/wear were evaluated as having the same score (grade 2) at the second and at the first follow-up evaluation. The criteria aesthetic anatomical form, marginal adaptation, tooth integrity, retention, and surface luster were all evaluated one grade higher at the second evaluation and therefore scored clinically good results (grade 2).

## Data Availability

The data are not publicly available due to privacy requirements.

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
