# Peer review of "A 23-Year Observational Follow-Up Clinical Evaluation of Direct Posterior Composite Restorations"

_dentistry, 2023, doi:10.3390/dj11030069_

Round 1

Reviewer 1 Report

Dear authors, thank you for submitting the manuscript entitled “Direct posterior composite restorations: a more than 23-year evaluation – an observational follow-up clinical study”.

The manuscript needs a major evaluation by a native English speaker because several sentences and phrases are not clear.

Rephrase the title, and put specific time either 23- or 24-years study.

Line 38: “tooth substance…” what is this ?

Line 40. Broad selection of composites for restorations in posterior teeth. I have classification of composites as packable, flowable and different amount of fillers but never a company offering only a composite for posterior teeth. Please provide more information giving examples with types and brands that offer composites only for posterior teeth.

Line 46-48. You claim hybrid, nanohybrid and nanofilled are gold standards, please describe in detail the specific differences of each one.

Line 52, if you mention a limited lifespan, then you have to be accurate regarding the time of some study, so include information what is limited for you (how long?).

Lines 52-60, in your third paragraph, you are making several statements about limited lifespan / longevity of restorations but you are not providing specific time longevity /lifespan of current studies. Provide specific information for your statements.

For the inclusion criteria please include information if patients were provided or used night-guard.

For the inclusion criteria, were the teeth in function / in occlusion?

For the inclusion criteria, did teeth have adjacent teeth with ideal contacts ?

I see a date for the second follow-up examination, but I do not see a date for the first follow-up in this study, so please provide it. If the first follow-up was placed back in 1995-96 then it was really the first follow-up for this study.

Major problem here is that there was no a periapical or bitewing radiographs in the follow-up. It is necessary radiographs for examinations, and it is not considered unnecessary radiation. You have to fully explain this and show other study where they did not take radiographs for follow-up. Otherwise, this is study has little value, because it did not follow minimum the exam requirements.

Give a detailed reason why they investigator did not use the 16 criteria. Provide examples of other studies were successfully published that did not follow the criteria as well.

You evaluated single and multi-surface composite restorations. Please mention if the results show significant difference in success among them.

Do not use the word “filling”; you need to use a more professional terminology such as direct restoration or simply restoration.

Reviewer 2 Report

This is an interesting study that investigated the clinical longevity of direct posterior composite restorations. Composite materials are widely used in restorative dentistry due to their esthetic characteristics and good mechanical properties. Today, the need for the posterior composite restorations is continuously increasing because of the high esthetic demands of the patients. Most clinical studies have focused on comparing new types of resin composites with observation times seldom exceeding 5 years, and few articles have investigated composite fillings survival in posterior teeth in the long term.

The major strengths of this study include the following:

1. The scope of the study is of a major interest to dental practitioners.

2. The paper is well written and documented.

3. The authors investigated long-term composite restorations survival rate.

I am strongly inclined to recommend this article for publication in the Dentistry Journal.  However, I have some suggestions to the authors:

1. English changes are required.

 2. I think it is better to present the information from the following paragraphs in a Flow Diagram (thus, it will be easier for the readers to perceive it):

“219 patients with a total of 329 composite posterior dental restorations fulfilled the inclusion criteria for the study. Out of the 219 patients, 108 patients lacked complete patients records regarding the clinical evaluation and did therefore drop out of this study. The remaining 111 patients were invited by letters, E-mails, and phone calls to visit the Carolinum Dental-University Institute. Out of diverse reasons patients dropped out of the study: there were no current correct contact details of them available (n=74), they did not accept the invitation (n=7), patients could not participate due to high age (n=4), severe illness (n=3) or death of patient (n=1). Correspondingly, 22 patients (13 male and 9 females, mean age 66.1 [range: 50 to 84] years) with a total of 42 restorations attended the follow- up examination.”

“Twenty-two out of initially 219 patients took part in the second follow-up examination after more than 23 years, of which 9 participants were women and 13 participants were men. 42 out of initially 329 restorations were examined. The distribution of the group variables of the 42 restorations was as followed: 20 restorations in the maxilla and 22 restorations in the mandible; 19 restorations in premolars and 23 restorations in molars; 14 single-surface fillings and 28 multi-surface fillings. From these 42 restorations that got examined a total of 16 were no longer in situ out of diverse reasons and therefore got examined with a grade 6: 5 failed due to missing tooth with corresponding restoration that should have gotten examined. From the 5 missing teeth, 2 had been replaced by implants and in 3 cases there was a consisting tooth gap. 37 teeth were still in situ, but of which already 3 had ceramic crowns, 2 had gold inlays, 1 had a partial gold crown and of which 31 had composite restorations, but 5 of these had already been replaced for a new composite restoration before the second follow-up examination took place. A total of 26 restorations were still in situ in 17 patients, which were evaluated using the selected FDI criteria.”

3. In my opinion, the Discussion section is too long and includes texts that do not explain the results but fit better in the Introduction section. I think it should be appropriately reduced and restricted to discussing the results of the present study.

Furthermore, there are some duplicates in the text:

Lines 62 – 79 in the Introduction section duplicate the information in the Discussion (Lines 457-486).

Lines 573 – 577: “van Dijken et al. and several other clinical findings stated, that there are probably more important factors influencing marginal adaptation and longevity of posterior composite restorations than the polymerization shrinkage stress of common composite materials, for example the operators technique as well as individual risk factors of the participants like caries risk and bruxing activity.” duplicate the following sentence:

Lines 623 – 627: “This is also described by van Dijken et al. [47] by saying that polymerization shrinkage is very likely not the most crucial factor for the endurance of resin composite restorations but factors like technique of the operator and patient related risk factors like parafunctional habits and caries activity and the margin of the restoration under oral cavity conditions [39] play a role.”

Reviewer 3 Report

Although the study is interesting. However, so many flaws make this study weak:

1. There seems no standardization in evalaution method. the selected population, i.e., 18-70 yrs would definately influence the results. The dietary habits of the person varies from age to age. A person in 20s would have a different dietary habits than a person in 50s or 60s.

2. All restorations were done on posterior teeth but even in posterior region masticatory load varies. !st molar bear different load than premolars and 2nd molar. Hence, the clinical life of a restoration is also affected.

3. The type of cavity (i.e., class I or Class II  etc) would also have different outcomes. There seems no standardization in data collection.

4. Survivability of restorations also depend on the cavity size and area.

5. All the evaulation is based on visual analysis. The outcome of qualitative analysis is difficult to generalize.

6. Sample size is also very small. The results can not be generalized

7. There are so many spelling mistakes through out the paper

8. The authors also need to improve the paper by scientifically and grammarly editing

Round 2

Reviewer 1 Report

Dear authors,

You have addressed all my questions and the quality of the manuscript has significantly improved. I recommend your article for publication.

Thanks.

Reviewer 3 Report

No more comments